# Morphological and hormonal diversity in rose (*Rosa hybrida* L.) and potato (*Solanum tuberosum* L.) Ri genotypes: A comparative study

Philipp Rüter[1‡], Tim Thomsen[2‡], Jarosław Tyburski[3], Piotr Waligórski[4], Ewa Surówka[4], Jacek Kęsy[3], Natalia Mucha[3], Manh Hung Doan[1], Traud Winkelmann[1*], Thomas Debener[2]

1 Section Reproduction and Development, Institute of Plant Genetics, Faculty of Natural Sciences, Leibniz University Hannover, Hannover, Germany, 2 Section Molecular Plant Breeding, Institute of Plant Genetics, Faculty of Natural Sciences, Leibniz University Hannover, Hannover, Germany, 3 Chair of Plant Physiology and Biotechnology, Institute of Biology, Faculty of Biological and Veterinary Sciences, Nicolaus Copernicus University in Toruń, Toruń, Poland, 4 The Franciszek Górski Institute of Plant Physiology, Polish Academy of Sciences, Kraków, Poland

‡ These authors share first authorship and contributed equally to this work.
* traud.winkelmann@zier.uni-hannover.de

## Abstract

Ri (root-inducing) technology, mediated by *Rhizobium rhizogenes*, presents a promising approach for modifying plant architecture. However, a comprehensive understanding of how complete, integrated wild-type T-DNA alters plant physiology is still lacking, as most related research has focused on partial gene sets and single species. To address this gap, our study undertakes the first systematic, comparative analysis of Ri genotypes across the phylogenetically distant species of rose (*Rosa hybrida* L.) and potato (*Solanum tuberosum* L.). We aimed to correlate organ-specific T-DNA gene expression with hormonal profiles and growth traits, thereby identifying both the general mechanisms of the Ri phenotype and species-specific differences. Morphological, molecular, and hormonal data were collected from three plant organs—leaves, stems, and roots—and analyzed across multiple Ri genotypes generated by *R. rhizogenes* strain ATCC 15834. All the Ri genotypes contained $T_L$ (left T-DNA) but differed in the presence of $T_R$ (right T-DNA) sequences. Compared with the respective wild-type plants, the Ri genotypes consistently presented shorter internodes (ratio of 0.5–0.89), in most cases smaller leaves (ratio of 0.52–1.03) and greater in vitro root formation (ratio of 0.71–2.34), as well as diverse expression levels of the *rol* and *aux2* genes, a consistently higher concentration of cytokinins and altered levels of stress-related hormones in specific organs. Correlation and principal component analyses confirmed the relationships among *rol* gene expression, root number, and reduced shoot growth, whereas hormone responses were species-specific. Together, these findings complement the characteristics of typical Ri plants shown in previous studies and provide new insights into the underlying hormonal and genetic mechanisms, indicating altered stress signaling pathways in Ri

which permits unrestricted use, distribution, and reproduction in any medium, provided the original author and source are credited.

**Data availability statement:** All relevant data are within the paper and its Supporting Information files.

**Funding:** This research was funded by the DFG (Deutsche Forschungsgemeinschaft, https://www.dfg.de/, WI 2002/6–1) (Authors: PR, TW, TD) and NCBR (National Centre for Research and Development, https://www.gov.pl/web/ncbr-en) (Authors: JT, JK, NM) as part of the RootsPlus project. RootsPlus was carried out under the second call of the ERANET Cofund SusCrop, being part of the Joint Programming Initiative on Agriculture, Food Security and Climate Change (Facce-JPI). SusCrop has received funding from the European Union's Horizon 2020 research and innovation program under grant agreement No 771134. The funders did not play any role in the study design, data collection and analysis, decision to publish or during preparation of this manuscript.

**Competing interests:** The authors have declared that no competing interests exist.

**Abbreviations:** ABA, Abscisic acid; ags, Agropine synthase; ARF, Auxin response factor; aux, Auxin biosynthesis gene; BAP, 6-Benzylaminopurine; cZ, cis-Zeatin; cZR, cis-Zeatin riboside; Ct, Threshold cycle; DHZ, Dihydrozeatin; DHZR, Dihydrozeatin riboside; Ds, 'Desiree'; DsRed, Discosoma striata red fluorescent protein; GA3, Gibberellic acid 3; gDNA, genomic DNA; GFP, Green fluorescent protein; IAA, Indole-3-acetic acid; IBA, Indole-3-butyric acid; iP, N6-(Δ²-Isopentenyl)adenine; IQR, Interquartile range; JA, Jasmonic acid; JAMe, Methyl jasmonate; KR, Kinetin riboside; mas1, Mannopine synthase; X, meta-Topolin; MS, Murashige and Skoog (1962); ND, 'New Dawn'; NTC, No template control; oT, ortho-Topolin; PC, Principal component; PCA, Principal component analysis; PCR, Polymerase chain reaction; PPFD, Photosynthetic photon flux density; Ri, Root inducing; rolA-D, Root oncogenic locus A-D; ROS, Reactive oxygen species; RQ, Relative quantity; RT-qPCR, Quantitative reverse transcription PCR; SA, Salicylic acid; T-DNA, Transfer DNA; $T_L$, Left T-DNA; $T_R$, Right T-DNA; tZ, trans-Zeatin; tZR, trans-Zeatin riboside.

genotypes. The first evaluations of root hair data in Ri plants contradict assumptions about a hairy root phenotype but also indicate an increased root tip diameter (ratio of 1.06–1.23) compared with that of the wild-type. Together, these findings underscore the importance of balancing architectural benefits with potential physiological trade-offs in the use of Ri technology for future breeding applications.

## Introduction

A biotechnological tool that has recently received increased attention offers an alternative approach for addressing breeding challenges [1–3]: the so-called Ri (root-inducing) technology uses natural soil-borne rhizogenic rhizobia such as *Rhizobium rhizogenes* (formerly *Agrobacterium rhizogenes*). This bacterium modifies plant cells by incorporating T-DNA (transfer DNA) from its Ri plasmid into the genome. When so-called *rol* (*root oncogenic locus*) genes on the T-DNA are expressed, they modify plant tissue hormone homeostasis, resulting in the development of "hairy roots" across a broad range of plant species [2]. These hairy roots can either be propagated on tissue culture media lacking plant growth regulators or can be forced to regenerate whole plants. The resulting Ri genotypes likewise carry integrated bacterial T-DNA in their genomes. Ri plants are generally not considered to be genetically modified organisms, as long as natural wild strains are being used [4].

The Ri plasmids of rhizogenic rhizobia contain several strain-specific open reading frames on their T-DNA [5], all of which potentially affect the morphology of Ri plants. Plasmids of types I and II contain one T-DNA, whereas plasmids of type III carry two separate types of T-DNA, called $T_L$ (left T-DNA) and $T_R$ (right T-DNA), which are transmitted independently of each other [6]. Among these fragments, only the $T_L$-DNA is essential for hairy root induction and carries the *rol* genes [7]. T-DNA genes in regenerated Ri genotypes affect plant morphology by altering hormone homeostasis [8]. Characteristic Ri traits include reduced height due to shortened internodes and smaller, wrinkled leaves. Additionally, stronger and more plagiotropic root growth and increased branching were observed, as well as altered flower quantity, shape, and timing of bloom [2].

The mechanisms through which T-DNA gene expression influences hormone status and plant architecture are still being studied. Several studies have investigated the effects of single *rol* genes on hormone profiles, providing insights into gibberellic acid, cytokinin and auxin homeostasis. However, these investigations rarely analyzed complete sets of T-DNA-encoded genes [9–20]. Moreover, previous research has focused primarily on single species, such as *Nicotiana benthamiana* or *Arabidopsis thaliana*, while cross-species studies remain limited. Different T-DNA gene expression levels rarely correlated with hormone profiles when different plant organs were considered. Furthermore, no reports exist concerning the root hair status of Ri genotypes, although *rol* genes are known to induce a "hairy" phenotype in roots formed through rhizogenic rhizobia transformation [21]. The hypothesis of enhanced root hair formation in Ri genotypes is further supported by the observations of Bose et al.

(2022) [22], who reported a *rolB*-induced upregulation of the genes *auxin response factor 7* (*ARF7*) and *ARF19*, which are also associated with root hair growth and root hair elongation.

The main objective of this study was to elucidate the fundamental differences between Ri and wild-type (non-Ri) genotypes by linking T-DNA gene expression to growth habits and hormone profiles through correlation analyses. The investigation included three distinct plant organs: leaves, stems and roots. Two phylogenetically and physiologically diverse species were selected for comparative analysis: *Rosa hybrida* L., a member of the Rosaceae family and the rosids clade, representing woody plants with two different cultivars, and *Solanum tuberosum* L.*,* an agriculturally important herbaceous crop from the Solanaceae family and the asterids clade. This study compared Ri genotypes containing only natural $T_L$ T-DNA sequences with other Ri genotypes containing both $T_L$ and $T_R$ T-DNA sequences. Additionally, the root hair status of rose Ri genotypes was examined, and potato tuber characteristics were evaluated.

## Materials and methods

### Plant material

In this study, the Ri genotypes of two species, members of the genus *Rosa* and the species *Solanum tuberosum*, were analyzed. Specifically, three genetic backgrounds were included: *Rosa* Am, *Rosa* 'New Dawn' (abbreviated ND), and *Solanum tuberosum* 'Desiree' (abbreviated Ds). *Rosa* Am is an experimental hybrid that resulted from crossing the cultivars 'AUSmas' and 'Pariser Charme'. The latter parent contributed the *muRdr1A* transgene, which is involved in the defense reaction to the fungal leaf pathogen *Diplocarpon rosae*, with the transformed rose genotypes showing increased resistance [23,24]. To date, no formal study has reported the influence of *muRdr1A* on plant morphology, with its function seemingly confined to disease resistance during the effector-triggered immune response. We therefore deemed it to be irrelevant for the current study. For convenience, all three genetic backgrounds are referred to as "cultivars". All three wild-types and their Ri genotypes are listed in Table 1.

**Table 1. Genotypes and their genetic constitution regarding the PCR-proven presence of the T-DNA genes, integrated by transformation with *R. rhizogenes* strain ATCC 15834.**

| Genus/species | Cultivar | Genotype | $T_L$ genes | $T_R$ genes | Reporter gene |
|---|---|---|---|---|---|
| *Rosa* | Am | Am (wild-type) | – | – | – |
| *Rosa* | Am | Am1 | *rolA, rolB, rolC, rolD* | – | *DsRed* |
| *Rosa* | Am | Am2 | *rolA, rolB, rolC, rolD* | – | *DsRed* |
| *Rosa* | Am | Am3 | *rolA, rolB, rolC, rolD* | – | *DsRed* |
| *Rosa* | Am | Am4 | *rolA, rolB, rolC, rolD* | *aux1, aux2, mas1, ags* | *DsRed* |
| *Rosa* | Am | Am5 | *rolA, rolB, rolC, rolD* | *aux1, aux2, mas1, ags* | – |
| *Rosa* | Am | Am6 | *rolA, rolB, rolC, rolD* | *aux1, aux2, mas1, ags* | – |
| *Rosa* | 'New Dawn' | ND (wild-type) | – | – | – |
| *Rosa* | 'New Dawn' | ND1 | *rolA, rolB, rolC, rolD* | *aux1, aux2, mas1, ags* | – |
| *Rosa* | 'New Dawn' | ND2 | *rolA, rolB, rolC, rolD* | *aux1, aux2, mas1, ags* | *GFP* |
| *S. tuberosum* | 'Desiree' | Ds (wild-type) | – | – | – |
| *S. tuberosum* | 'Desiree' | Ds_noRi | – | – | *GFP* |
| *S. tuberosum* | 'Desiree' | Ds1 | *rolA, rolB, rolC, rolD* | – | *DsRed* |
| *S. tuberosum* | 'Desiree' | Ds2 | *rolA, rolB, rolC, rolD* | – | *GFP* |
| *S. tuberosum* | 'Desiree' | Ds3 | *rolA, rolB, rolC, rolD* | *aux1, aux2, mas1, ags* | *GFP* |
| *S. tuberosum* | 'Desiree' | Ds4 | *rolA, rolB, rolC, rolD* | *mas1, ags* | *GFP* |

Ds_noRi = regenerated, but not containing T-DNA genes. $T_L$ = left T-DNA, $T_R$ = right T-DNA, *rolA-D* = root oncogenic locus A-D, *aux1-2* = auxin biosynthesis genes, *mas1* = mannopine synthase, *ags* = agropine synthase, *DsRed* = Discosoma striata red fluorescent protein, *GFP* = green fluorescent protein.

The Ri genotypes were derived from transformation with the *R. rhizogenes* strain ATCC 15834 [25], which was provided by Dr. Frank Dunemann (Julius Kühn Institute, Quedlinburg, Germany). The strain was used either as a wild-type strain or carrying a binary plasmid with either a *GFP* (C757 pGFP U10) or *DsRed* (B469 pRed U10) reporter gene. The plasmid maps can be downloaded from the DNA Cloning Service (Hamburg, Germany) website. Additionally, one Ds genotype (Ds_noRi) was included, which was regenerated from a GFP-transformed root lacking *rol* genes, and served as a regeneration control.

The transformation and regeneration procedures for the rose Ri genotypes are described in supplementary S3 File Materials and Methods. Induction of hairy roots on potato shoots and the regeneration of Ri plants were performed according to Visser et al. (1989) [26]. The verification of the integration of the $T_L$ and $T_R$ genes of all the Ri genotypes was performed via PCR and gel electrophoresis by Thomsen (2024) [27], and the protocols are described in Rüter et al. (2024) [3].

The plants were subsequently grown in different batches for morphological characterization and gene expression and hormone analysis (overview in S3 File Materials and Methods, Table A). Detailed information about each organ and type of analysis is provided in the Materials and Methods section.

## Morphological analysis

**In vitro root measurement.**  Three-week-old in vitro propagated shoots were used for the rooting experiments, which were carried out twice. Rose shoots were cultured on medium containing 1 MS (Murashige and Skoog, 1962 [28]) salts supplemented with 100 mg/L FeEDDHA substituted for FeNaEDTA, 1 MS vitamins, 3% sucrose, 2.22 μM BAP, 0.06 μM $GA_3$, and 8 g $L^{-1}$ plant agar (Duchefa, Haarlem, The Netherlands) at pH 5.8. The cultures were maintained at 24 °C. Potato plants were propagated on medium consisting of 1 MS salts and 1 MS vitamins, 3% sucrose and 8 g $L^{-1}$ plant agar at a pH of 5.8 and maintained at 18 °C and a photosynthetic photon flux density (PPFD) of 55 μmol $m^{-2}$ $s^{-1}$. Following propagation, rose shoots were rooted on vertical plates containing ½ MS salts and vitamins, 3% sucrose and 8 g $L^{-1}$ agar at pH 5.8. The Am cultivar and its Ri genotypes received an additional 0.4 μM IBA because of its reduced rooting capacity. Potato shoots were rooted on the same medium used for propagation, except that 6 g $L^{-1}$ plant agar was used. One hundred mL of medium was poured into 12 x 12 x 1.7 cm vertical square plates (Carl Roth GmbH + Co. KG, Karlsruhe, Germany). The top quarter of the medium was removed, and in vitro shoots measuring 1.2–2 cm were placed on the cut surface of the medium. Three cuttings per plate and a total of three plates were used for rose, while one cutting per plate and a total of six plates were prepared for potato. The plates were positioned vertically under the same temperature and light conditions as before. Data acquisition involved capturing images of vertical plates after four weeks for rose and after two weeks for potato using a scanner (Perfection V800 Photo, Epson Deutschland GmbH, Düsseldorf, Germany). The root length was measured manually using Fiji version 2.16.0 software with the segmented line tool and the measure function. Additionally, the number of roots per plant was counted. The average root length was calculated by dividing the total root length by the number of roots. Data are recorded in S1 File Raw Data, in the "In_vitro_roots" sheet.

## Ex vitro growth parameter measurement

In vitro rooted shoots of rose and potato plants were transplanted into 7-cm pots containing peat substrate (type ED73: 70% peat H2-H5 and 30% clay, 340 mg/L N + 260 mg/L $P_2O_5$ + 330 mg/L $K_2O$ + 100 mg/L Mg; Einheitserde Werkverband e.V., Sinntal-Altengronau, Germany). The plants were acclimatized at 24 °C for three weeks under 100% humidity conditions and subsequently transferred to a room at 21 °C under a 16-hour light and 8-hour dark cycle with 100 μmol $m^{-2}$ $s^{-1}$ PPFD. After one additional week, the humidity-maintaining cover was removed from the plants. One week later, the plants were transferred to 14-cm pots and supplemented with 5 g of slow-release fertilizer pellets (Floranid® Twin Permanent 16N-7P-15K; COMPO EXPERT GmbH, Münster, Germany). After five more weeks of growth, three plants per genotype were phenotyped. The shoot number per plant was counted, and internode length measurements were taken from the

15 uppermost internodes starting from the shoot tip of each plant. These data are recorded in S1 File Raw Data, in the "In_vivo_internodes" sheet. Leaf parameters, including the compound leaf length, leaflet length, and leaflet width, were measured for the 8 uppermost unfolded leaves per rose plant and 10 leaves per potato plant. These measurements are recorded in the sheet "In_vivo_leaves". For the Ds genotypes, the tubers were harvested, counted, and weighed after a total of 16 weeks of growth and the results are listed in the "In_vivo_tubers" sheet. On the basis of the results of the morphological data analysis, three rose Ri genotypes with strong Ri traits (Am4, ND1, and ND2) were selected along with their respective wild-types for dry mass and root hair status evaluation. Details of the plant cultivation and root hair measurements are described in supplementary S3 File Materials and Methods.

## Statistical analysis of morphological data

Statistical analysis was performed with R version 4.4.2 in RStudio version 2024.12.0.467 using Dunnett tests, implemented in the emmeans package version 1.10.5 [29], with appropriate statistical models, where Ri genotypes were compared to their respective wild-types. The statistical models were fitted individually for each parameter and cultivar. Linear (mixed) models were applied for metric data, whereas generalized linear models with quasi-Poisson distributions were utilized for counting data. Data transformation was performed using the BestNormalize version 1.9.1 package [30] on the basis of residual distribution patterns, utilizing the DHARMa package version 0.4.7 [31]. The model formulas, distributions, and transformation types are documented in file S2 File Data Summary, in the "Statistical models" sheet. A heatmap was created using the pheatmap package version 1.0.12 [32]. In the heatmap the ratios are displayed, which were calculated by dividing the mean of the respective parameter of each Ri genotype by the mean of the respective wild-type. The plots of the tuber and root hair data were created with ggplot2 version 3.5.1 [33]. The plant photographs and the heatmaps were arranged and adjusted using Affinity Designer version 1.10.6.1665 (Serif (Europe) Ltd., Nottingham, United Kingdom).

## T-DNA gene expression analysis

For the gene expression analysis, samples were taken from in vitro roots after four weeks on rooting medium. Leaf and stem (internodes and nodes) samples were taken from plants three weeks after acclimatization. Samples of three independent biological replicates (plants) were frozen in liquid nitrogen and stored at −80 °C. After homogenization, RNA was extracted using the Quick-RNA Miniprep-Kit (Zymo Research Europe GmbH, Freiburg, Germany), and gDNA was digested with DNase I. cDNA synthesis was performed with 300 ng of RNA using the LunaScript RT SuperMix Kit, which contained random hexamer and oligo-dT primers (New England Biolabs, Frankfurt am Main, Germany). Contamination with genomic DNA (gDNA) was checked via PCR using intron-spanning actin primers for rose from Hattendorf and Debener (2007) [34] or those for potato from Saubeau et al. (2016) [35] (supplementary S3 File Materials and Methods, Table B), running at 94 °C for 3 min, followed by 35 cycles of 95 °C, 55 °C and 72 °C, each for 30 sec, and a final extension at 72 °C for 10 min. PCR products were loaded on a 2% agarose gel with a DNA ladder (Thermo Scientific™ GeneRuler 100 bp from Fisher Scientific) and run at 5 V/cm for 40 minutes. The primer efficiencies for quantitative reverse transcription PCR (RT–qPCR) were determined for *rolA*, *rolB*, *rolC* and *aux2*. The amplification specificity of each primer pair in the experimental samples was verified by agarose gel electrophoresis with PCR products using the abovementioned protocol. The sequences were obtained from Lütken et al. (2012b) [36] and are listed alongside their efficiencies and fragment lengths in supplementary file S3 Materials and Methods, in Table C. The dilution series (1:1, 1:4, 1:16, 1:64, 1:256, 1:1024, 1:4096) were prepared from mixed samples of the Am genotypes. Three technical replicates were tested for each dilution step. All RT–qPCR analyses were performed in hard-shell 384-well PCR microtiter plates (Bio-Rad Laboratories GmbH, Feldkirchen, Germany) with a total reaction volume of 8 µL. Luna Universal qPCR Mix (New England Biolabs, Frankfurt am Main, Germany) was mixed with 30 ng of cDNA and 200 nM of the respective forward and reverse primers. Respective no template controls (NTCs) were also added to each primer pair. The microtiter plates were sealed

with protective transparent foil (adhesive sealing film; nerbe plus GmbH & Co. KG, Winsen/Luhe, Germany) and centrifuged at 4000 rcf for 10 min. The plates were then placed into a QuantStudio 6 Flex Real-Time-PCR System (Thermo Fisher Scientific, Waltham, United States). The following protocol was used for amplification: 1 min at 95 °C, followed by 40 cycles of 15 s at 95 °C and 30 s at 60 °C. The amplification specificity for each primer pair was assessed by a melting curve analysis ranging from 60 °C to 95 °C with temperature increments of 0.5 °C after each RT–qPCR run during both the primer testing and expression analysis. The data were analyzed using QuantStudio™ Real-Time PCR Software (Applied Biosystems, version 1.3) to determine the efficiency and $R^2$. The efficiencies of the reference genes for potato (*OXA1* and *RPN7*, reported by Castro-Quezada et al. (2013) [37]) and rose (*SAND* and *TIP*, reported by Klie and Debener (2011) [38]) have previously been evaluated. All Ct (threshold cycle) values are documented in file S1 Raw Data, in the "Expression_data" sheet. The mean Ct values of three technical replicates were corrected within RStudio by the respective primer efficiency with the following formula:

$$expression\ level = \left( \frac{efficiency\ [\%]}{100} + 1 \right)^{(-mean(Ct\ technical\ replicates))}$$

The relative quantity (RQ) was calculated by dividing the expression level of T-DNA genes by the geometric mean of both reference genes for each biological replicate. OrderNorm transformation of RQ values was applied for statistical analysis with linear models with genotype as a fixed effect for each cultivar, organ and gene separately. Afterward, ANOVA was conducted, followed by Tukey's post hoc test, using the emmeans package for genotype comparisons.

## Plant hormone analysis

In vitro generated roots were collected after four weeks on rooting medium, while leaves and stems (internodes together with nodes) were harvested three weeks after acclimatization. Fresh mass samples (100−200 mg per sample) were prepared from 4–7 independent biological replicates (plants) per genotype and subsequently frozen in liquid nitrogen before being stored at −80 °C and lyophilized for three days. The concentration of plant hormones was determined by liquid chromatography–tandem mass spectrometry (LC–MS/MS) using internal deuterated standards.

The following plant hormones were analyzed: the auxin indole-3-acetic acid (IAA), the cytokinins cZR (cis-zeatin riboside), tZR (trans-zeatin riboside), DHZR (dihydrozeatin riboside), iP (Δ²-isopentenyl adenine), KR (kinetin riboside), mT (meta-topolin), oT (ortho-topolin), cZ (cis-zeatin), tZ (trans-zeatin) and DHZ (dihydrozeatin), and the stress hormones ABA (abscisic acid), SA (salicylic acid), JA (jasmonic acid) and JAMe (methyl jasmonate). The procedure details and detection limits are provided in the supplementary Materials and Methods. Gibberellic acids were not analyzed despite their potential relevance for Ri plants. However, since GAs are a large hormone family, a separate study should be conducted in which GA contents are correlated with the growth patterns of different organs of Ri plants. Given that the concentrations of several GAs are rather low in plant tissues, equipment with adequate sensitivity and sufficient sample mass is necessary.

Plant hormone data were obtained from two independent laboratories, which used either a Nexera XR UHPLC (Shimadzu, Kyoto, Japan) coupled to an LCMS-8045 (Shimadzu, Kyoto, Japan) or an Agilent 1260 HPLC system coupled to an Agilent 6410 Triple Quadrupole mass spectrometer (see supplementary S3 File Materials and Methods for more details). However, we considered it justified to analyze them jointly, as the interpretation was solely based on relative differences (fold changes in concentration in Ri genotypes and the respective wild-types, as explained below) rather than absolute concentrations, and no direct comparisons were made between datasets generated by different analytical methods. Moreover, the levels of major plant hormones were generally consistent between the two groups. Furthermore, not all the cultivars and organs could be analyzed for all the hormones; e.g., the ND Ri genotypes were not available for root hormone analysis. The complete matrix indicating which hormones were analyzed in which samples is provided in S1 File Raw Data, in the "Analyzed_hormones" sheet. Measurements of successfully detected hormones are recorded in the "Hormone_data" sheet.

To ensure a valid analysis, outlier filtering was applied to all the analyzed hormones. Outliers were removed using the interquartile range (IQR) criterion, where the IQR was calculated as the upper quartile minus the lower quartile, and bounds were set at the upper or lower quartile $\pm 5 \times$ IQR. Instead of calculating ratios as performed for the morphological data, log2-fold changes (log2(x + 1)) were calculated for the hormone concentrations in the Ri genotypes compared with those in the wild-type plants using the following formula:

$$log2 \text{ fold change} = log2(Ri \text{ genotype mean} + 1) - log2(wildtype \text{ mean} + 1)$$

This approach enabled a better visual comparison of different hormones in the heatmap. For statistical evaluation, the hormone concentrations of each sample were likewise log2(x + 1)-transformed, and a linear model was fitted for each cultivar and organ separately, with the genotype as a fixed effect. Dunnett's test with the emmeans package was subsequently applied to compare Ri genotypes to their respective wild-types. A heatmap was created using the pheatmap package and adjusted using Affinity Designer.

### Correlation analyses

Correlations were calculated with the respective parameters for each organ separately (e.g., leaf width with IAA in leaves) with one value (based on means) per genotype. The analysis was performed in RStudio using the cor.test function with Spearman correlation, as most data were not normally distributed, and values based on the means of genotypes represented single rather than multiple measurements. Additionally, nonlinear patterns were expected because of the small number of values and different measurement scales. Correlations were first calculated for all the cultivars together by standardizing across the cultivars with ratios of genotype means (Ri genotype/wild-type) for the morphological data and log2-fold changes for the hormone data. For the gene expression data, the mean RQ for each Ri genotype was scaled between 0.1 and 1 across the rose and potato species per gene and organ using the following formula:

$$scaled \text{ value} = 0.1 + (value - min\_value) \times 0.9 / (max\_value - min\_value)$$

Only one wild-type (one data point) was used for the correlations because all three wild-types possessed the same values (1 for morphological ratios, 0 for hormone log2-fold changes, and 0 for scaled expression levels). Correlations were also calculated for each cultivar separately to identify similar or contrasting trends. For this analysis, the raw means were used, and 0 was used for the wild-type expression data, which was possible because of the use of the Spearman correlation. All ratios and raw means used for the correlation analyses are listed in S2 File Data Summary.

### Principal component analysis

For principal component analysis (PCA), the same matrix was employed as for the correlation analysis, which included all the cultivars. However, the *aux2* gene expression as well as the tuber, dry mass and hairy root data were excluded from the PCA because missing values for the Ri genotypes were problematic for the principal component (PC) calculation. PCA was performed using the "principal" function from the "psych" package version 2.5.3 [39], and the resulting biplot was visualized using ggplot. Final adjustments to the color schemes and text positioning were made in Affinity Designer.

### Results

### Phenotypes of Ri genotypes

Distinct morphological differences were observed between the Ri genotypes and the respective wild-type plants. Compared with the respective wild-type plants, all the Ri genotypes exhibited stronger root growth under in vitro conditions, with generally greater root numbers and longer root lengths (Fig 1). The hairy root phenotype (e.g., greater numbers of root hairs) was

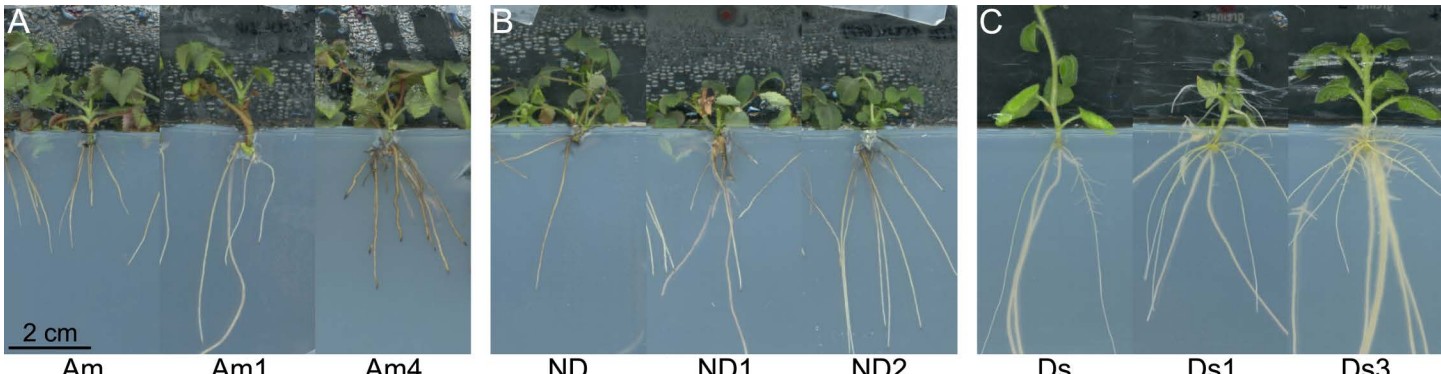

**Fig 1. Representative photos of in vitro rooted shoots of wild-types and two Ri genotypes per cultivar.** The photos were taken four weeks after root induction for roses and two weeks for potatoes. Left to right: wild-type, less pronounced Ri traits, stronger pronounced Ri traits. **(A)** *Rosa* Am. **(B)** *Rosa* ND. **(C)** *S. tuberosum* Ds.

observed only for roots of the potato Ri genotype Ds3 (Fig 1C). Similarly, typical aboveground Ri traits, including shorter internodes and smaller leaves, were observed after 10 weeks under greenhouse conditions (Fig 2). Yellowing of leaves was also observed in some Ri genotypes. This was particularly evident in the Ri genotypes Am1 and Ds3. In contrast, a darker leaf color was observed in the other genotypes, such as Am4.

Measurements of several morphological parameters revealed general patterns in most Ri genotypes across all the cultivars compared to the respective wild-type (Fig 3). Shorter internodes and lower shoot numbers were observed in some Ri genotypes. The leaves were smaller, and the roots were longer and more numerous. No consistent patterns were found when comparing plants with only $T_L$-DNA and plants with additional $T_R$-DNA integration. Means with SDs, n numbers and *p* values are provided in file S2 File Data Summary, in the "Morphological data" sheet; their graphical representations are presented in S4 File Results, in Fig A. When the regeneration control Ds_noRi without T-DNA genes was compared to the wild-type Ds, no significant differences in any morphological parameter were found, except for the in vitro root length, whose ratio was 0.58 to that of the wild-type (*p* value of 0.0154).

Tuber data and photographs are presented in S4 File Results, in Figs B and C. Tuber mass was significantly lower in Ds1 (2.1 g ± 2.47 g), Ds2 (2.89 g ± 2.18 g), and Ds4 (4.52 g ± 5.56 g) than in the control Ds (11.01 g ± 8.82 g). No significant differences in tuber number were detected.

On the basis of the results of the morphological data analysis, three rose Ri genotypes with strong Ri traits (Am4, ND1, and ND2) along with their respective wild-types were selected for investigation in a greenhouse experiment to assess root and shoot dry mass as well as root hair parameters. Dry mass measurements are documented in file S2 File Data Summary. Compared with the respective wild-type plants, the Ri genotypes presented reduced root and shoot dry masses. Am4, ND1 and ND2 presented shoot dry mass ratios to those of the wild-type of 0.51, 0.6, and 0.81, respectively. All the shoot dry mass reductions were statistically significant ($p < 0.001$). The root dry mass ratios were 0.9 for Am4, 0.62 for ND1, and 0.68 for ND2; the reductions in ND1 and ND2 were both significant ($p < 0.001$). The root hair data are shown in S4 File Results, in Fig D. No conclusive patterns were identified for the root hair number, length, or area measurements. Although significant differences were detected, they remained inconsistent across all Ri genotypes. Only the root body area was consistently and significantly elevated in all three Ri genotypes. Thus, the root hair measurements confirmed the visual impression that rose Ri genotypes did not produce roots with the characteristic hairy root phenotype.

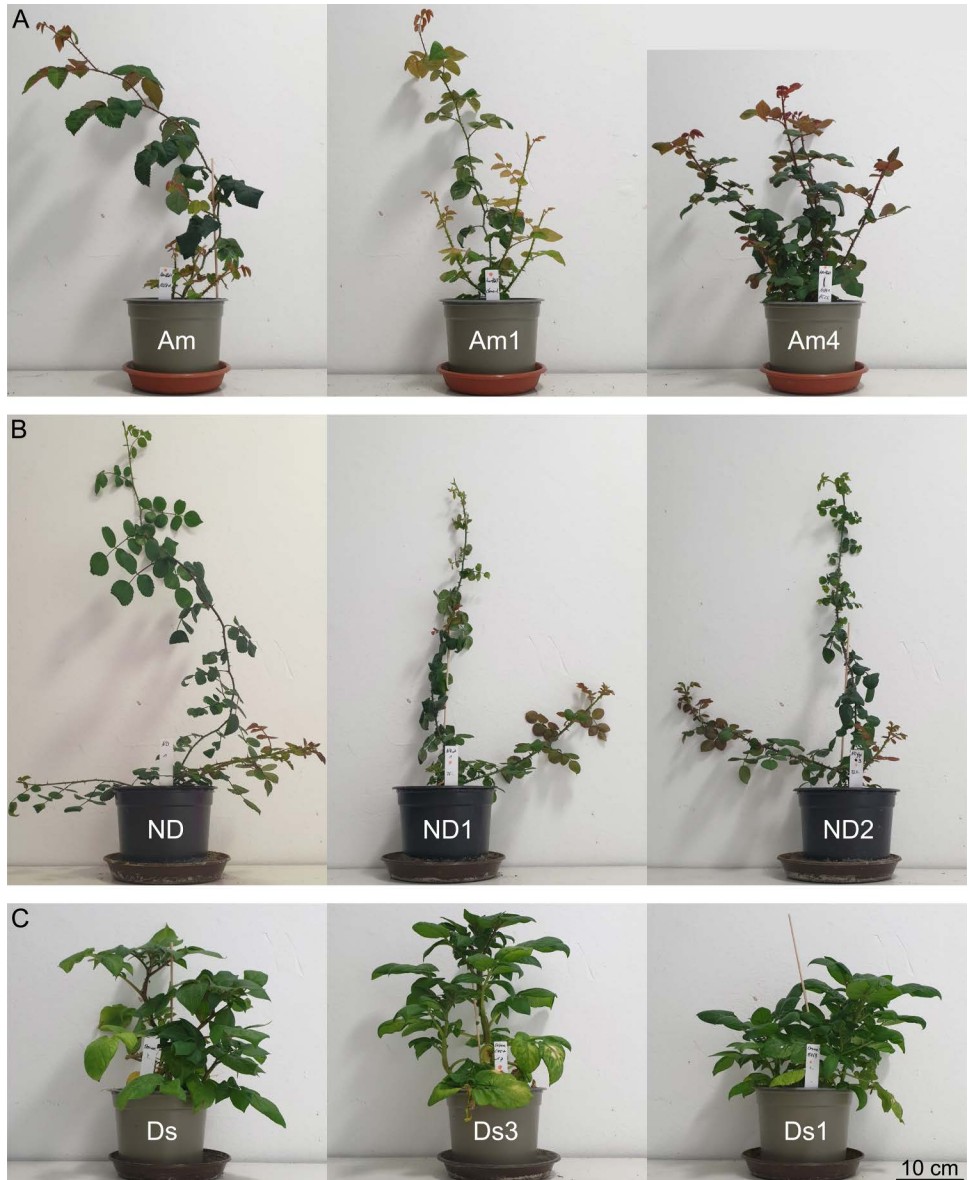

**Fig 2. Representative photos of greenhouse-grown wild-types and two Ri genotypes per cultivar, 10 weeks after acclimatization.** Left to right: wild-type, less pronounced Ri traits, stronger pronounced Ri traits. **(A)** *Rosa* Am. **(B)** *Rosa* ND. **(C)** *S. tuberosum* Ds.

## Expression of T-DNA genes

Greenhouse leaf and stem samples and in vitro root samples were collected from all the Ri genotypes for RT–qPCR analysis. The relative expression levels of the T-DNA genes *rolA*, *rolB*, *rolC*, and *aux2* were calculated and are presented in file S2 File Data Summary, within the "Expression data" sheet. The gene *aux2* was analyzed only for the rose genotypes with $T_R$ integrations (Am4, Am5, Am6, ND1, ND2). Although Ds3 also contains *aux2*, it was excluded because it is the sole *S. tuberosum* genotype with this gene, preventing statistical comparison within the cultivar. The expression of all selected T-DNA genes was demonstrated, with several Ri genotypes displaying significantly different expression levels of T-DNA

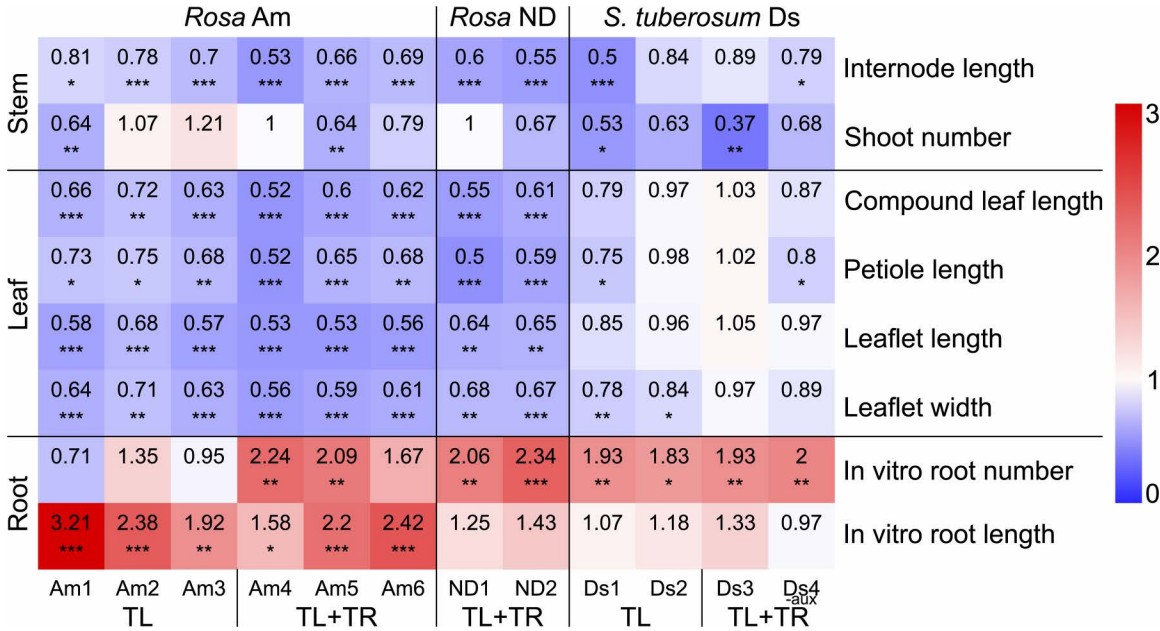

**Fig 3. Heatmap of morphological parameters of Ri genotypes compared to their respective wild-types.** Shown are ratios of the mean of the Ri genotype to the mean of the respective wild-type. Differences between means of Ri genotypes and wild-types were evaluated for significance with Dunnett's test, utilizing regression models, which are described in the Materials and Methods section. Significance thresholds are $p < 0.05 = *$, $p < 0.01 = **$ and $p < 0.001 = ***$.

genes across all organs and cultivars (ANOVA results are in Table A in S4 File Results; graphical representations of the relative expression levels are presented in S4 Fig in Fig E). Specifically, significant differences in expression among the Ri genotypes were most prominent for *rolA* and *rolB* in the stem and root tissues of the Am and Ds cultivars. In contrast, the Ri genotypes of ND displayed more uniform *rol* gene expression levels, with significant variation observed only for *aux2* in the leaves. The *rolC* gene showed the least variability, with significant differences among genotypes detected only in the leaves and stems of the Ds Ri genotypes. Graphical comparisons across species were deemed difficult to interpret because of the different reference genes used for the rose and potato species, resulting in incomparable relative expression between the species. Additionally, in contrast to the results of the analysis of the morphological and hormone data, no ratios or log2-fold changes could be calculated for the gene expression data, as there were no measurements for the wild-type plants.

## Plant hormone profiles

Leaf, stem, and in vitro root samples were collected from all Ri genotypes for plant hormone analysis. The hormone concentrations of the Ri genotypes were compared to those of the wild-type plants and are presented as log2-fold changes in the heatmap in Fig 4. The results revealed a consistent pattern of increased concentrations of the cytokinins DHZR and tZR in the stems of all the Ri genotypes. In the roots, the ratios of IAA and JA were mostly positive, as were those of ABA in the stems. Interestingly, contrasting patterns were observed between the two species rose and potato in leaf and stem tissues. In these organs, iP and IAA were present at lower concentrations in rose but at higher concentrations in potato Ri genotypes. The two rose cultivars also differed in terms of the ratios of certain hormones: the level of JA increased in the leaves of the ND Ri genotypes but decreased in the Am Ri genotypes. All means with standard deviations, sample numbers, *p* values, and log2-fold changes compared with those of the wild-type are provided in file S2 File Data Summary, in the "Hormone data" sheet, while graphical presentations are provided in S4 Fig Results, in Figs F-H.

|  | Rosa Am | | | | | | Rosa ND | | S. tuberosum Ds | | | | |
|---|---|---|---|---|---|---|---|---|---|---|---|---|---|
| **Leaf** | -0.4 | -1.77** | -1.29* | -1.44* | -0.15 | -0.73 | -0.14 | -0.14 | 0.41 | 0.21 | 0.27 | 0.23 | IAA |
|  | 1.33* | -1.7*** | -0.76 | 0.21 | -0.59 | -0.49 | 1.62*** | 1.13*** | 1.27* | 0.69 | 1.63** | 0.77 | tZR |
|  | 1.2*** | 0.98** | 0.63 | 0.57 | 1.06** | 1** | 1.83*** | 1.45*** | -0.5 | -0.53 | 0.19 | -0.32 | DHZR |
|  | -0.54 | -0.42 | -0.7 | 4.39** | -1.3 | 0.21 | -0.38 | 0.1 | -0.45 | 1.28** | 1.48** | 0.51 | cZR |
|  | -2.32** | -1.7** | -2.13** | 0.63 | -2.43*** | -2.39** | -0.02 | -0.96*** | 0.48* | 0.67** | 0.69** | 0.58** | iP |
|  | 1.65 | 2.13 | -0.04 | 1.94 | 0.86 | 0.83 | -1.27* | -1.36* | -0.39 | -0.38 | 0.28 | 0 | SA |
|  | -1.03* | -1.07* | -0.48 | -3.04*** | -0.49 | -0.42 | 1.15** | 1.34** | 1.06 | 0.26 | 0.39 | -0.12 | JA |
|  | -1.67* | 0.31 | -2.09** | 1.33 | -1.13 | -0.77 | 0.28 | 0.21 | -0.05 | -0.17 | -0.2 | -0.72 | ABA |
| **Stem** | 0.34 | -0.94 | -0.06 | -2.93*** | -0.27 | 0.26 | -0.66* | -0.12 | 0.43 | 0.19 | 0.27 | 0.25 | IAA |
|  | 2.08* | 1.95* | 1.88 | 0.76 | 1.92* | 3.09** | 2.74*** | 2.69*** | 0.42 | 1.39 | 0.55 | 1.4 | tZR |
|  | 2.92*** | 2.64** | 1.45 | 1.93* | 2.07* | 3.37* | 1.82*** | 1.96*** | 0.08 | 1.18 | 0.67 | 0.97 | DHZR |
|  | -0.15 | 0.49 | 0.41 | -0.35 | -0.86 | -0.27 | 0.04 | 0.26 | 0.17 | 1.19 | -0.21 | -0.56 | cZR |
|  | -1.08*** | -0.73* | -1.15*** | -0.73* | -1.2*** | -1.05** | -1.28*** | -1.46*** | 0.83 | 0.74 | 0.38 | 0.06 | iP |
|  | 0.22 | 1.91* | 2.58** | -0.93 | 0.37 | 0.36 | 0.02 | -0.7 | -1.03 | -1.4 | -1.23 | -1.38 | SA |
|  | 1.4 | 1.4 | 0.93 | -0.48 | 1.12 | 1.97 | -0.26 | -1.76** | -2.04 | -3.29** | -1.73 | -2.11 | JA |
|  | 0.4 | 1.59 | 0.76 | 1.32 | 0.35 | 1 | 0.12 | 0.85 | 0.7 | 0.4 | 0.92 | -0.08 | ABA |
| **Root** | -0.28 | 0.76 | 4.2* | 0.08 | 1.32 | 1.13 | NA | NA | 3.35** | 1.67 | 3.72*** | 2.72** | IAA |
|  | -0.61 | 1.2 | 1.63** | 1.7** | 1.08 | 0.84 | NA | NA | -0.08 | -0.12 | -0.11 | -0.21** | iP |
|  | -0.8 | -0.54 | 0.78 | 0.62 | 0.24 | -0.17 | NA | NA | 2.6*** | 0.88 | 1.83*** | 1.54** | SA |
|  | 1.69 | 1.71 | 1.37 | 0 | 1.7 | 2.63 | NA | NA | 2.24 | 3.58* | 4.13** | 3.03* | JA |
|  | -1.38 | 0.44 | 1.09 | 0.14 | 0.28 | 0.39 | NA | NA | 0.03* | -0.24 | -0.69 | 0.03* | JAMe |
|  | 0.83 | -0.23 | 0.1 | 0.95 | -1.17 | -1.94 | NA | NA | -0.2 | -1.18 | -2.47*** | -2.04*** | ABA |
|  | Am1 | Am2 | Am3 | Am4 | Am5 | Am6 | ND1 | ND2 | Ds1 | Ds2 | Ds3 | Ds4-aux | |
|  | TL | | | TL+TR | | | TL+TR | | TL | | TL+TR | | |

**Fig 4. Heatmap of plant hormone measurements in Ri genotypes relative to their respective wild-types.** Values depict log2-fold changes of the mean of the Ri genotype compared to the mean of the respective wild-type (0 = equal means of Ri and wild-type genotype. Differences between means of Ri genotypes and wild-types were evaluated for significance with Dunnett's test, utilizing linear regression models for each hormone, organ and cultivar with genotype as fixed effect and log2(x + 1)-transformed measurements. Significance thresholds are $p < 0.05 = *$, $p < 0.01 = **$ and $p < 0.001 = ***$. NA = not analyzed.

## Correlation analysis

A correlation analysis was performed on all the morphological, gene expression, and hormone data. Data from all the cultivars were analyzed together to identify meaningful insights into the general reactions of the Ri genotypes. Insights into

the correlations of T-DNA gene expression with hormonal and morphological parameters were the most important results, which are presented in Table 2. Other correlations, such as the correlation of *rolA* expression with *rolB* or the contents of IAA with JA, are reported in supplementary S4 File Results in Table B. Significant correlations were observed for *rol* gene expression with the root number (positive) and internode length (negative). Additionally, *aux2* expression was significantly negatively correlated with the leaflet size and IAA content in leaves. Further significant correlations are presented in the same table, in which all *rol* genes are shown to be positively correlated with each other in all organs. DHZR was negatively correlated with several leaf parameters, whereas iP and IAA were positively correlated. SA was negatively correlated with the root length but positively correlated with the shoot number. On the other hand, IAA was negatively correlated with the shoot number.

Correlation analyses were also conducted separately for each cultivar and are listed in S4 File Results, as an alternative approach to identify general (Table D) and cultivar/species-dependent correlations (Table C). As noted in Table C, JA in stems was positively correlated with the cytokinins DHZR and tZR in the Am cultivar but negatively correlated in ND and Ds. General correlations, i.e., correlations in the same direction across all three cultivars, confirmed that all *rol* genes were correlated with the leaf size, root number and internode length, as well as with each other. Positive correlation trends for *rol* genes with several hormones, such as ABA, DHZR and tZR, were identified for all the cultivars, especially in stems. Additionally, internode length was positively correlated with the shoot number.

## Principal component analysis

While correlation analysis revealed relationships between individual parameters, PCA provided a complementary approach by identifying underlying patterns and reducing the dimensionality of the dataset. The PCs in Fig 5 accounted for 34.4% (PC1) and 21.8% (PC2) of the total variance. All *rol* genes clustered together with the root number. A cluster of internode and leaf parameters appeared on the opposite side of PC2, indicating that smaller internodes were correlated with smaller leaf sizes and that both were negatively correlated with *rol* gene expression. Similarly, groups of several cytokinins are located in the positive PC2 and negative PC1 area, including DHZR and tZR. These cytokinins are orthogonal to *rol* gene expression, but they clustered together with the shoot number. Interestingly, only the auxin IAA in all three organs was on the positive side of PC1, which is in direct opposition to the in vitro root length and shoot number. The

**Table 2. Significant correlations of T-DNA gene expression with morphological and hormonal trait ratios relative to the respective wild-type across all rose and potato Ri genotypes.**

| Comparison | Parameter 1 | Parameter 2 | Organ | r | p | n |
|---|---|---|---|---|---|---|
| Gene expression - Morphology | *rolA* | In vitro root number | Root | 0.63 | 0.022 | 13 |
| | *rolB* | In vitro root number | Root | 0.66 | 0.013 | 13 |
| | *rolC* | In vitro root number | Root | 0.61 | 0.027 | 13 |
| | *rolC* | Internode length [mm] | Stem | −0.64 | 0.018 | 13 |
| | *aux2* | Leaflet length [mm] | Leaf | −0.84 | 0.036 | 6 |
| | *aux2* | Leaflet width [mm] | Leaf | −0.94 | 0.017 | 6 |
| Gene expression - Hormones | *rolC* | JA | Stem | −0.58 | 0.04 | 13 |
| | *aux2* | IAA | Leaf | −0.99 | 3.09e-04 | 6 |

Correlations of gene expression with expression of other genes, hormones with other hormones or morphological data are reported in the supplementary S4 File Results in Table B. Separate Spearman correlation analyses were performed at a significance threshold of 0.05 for the organs root (brown), stem (yellow) and leaf (green). Positive correlations are shown in blue, negative in red. For morphological data, simple ratios of means per genotype were used (1 for wild-types), for hormonal data the log2-fold change of means (0 for wild-types). For expression data, the mean RQ value was scaled between 0.1 and 1 (0 for wild-types). r = Spearman correlation coefficient, n = number of genotypes.

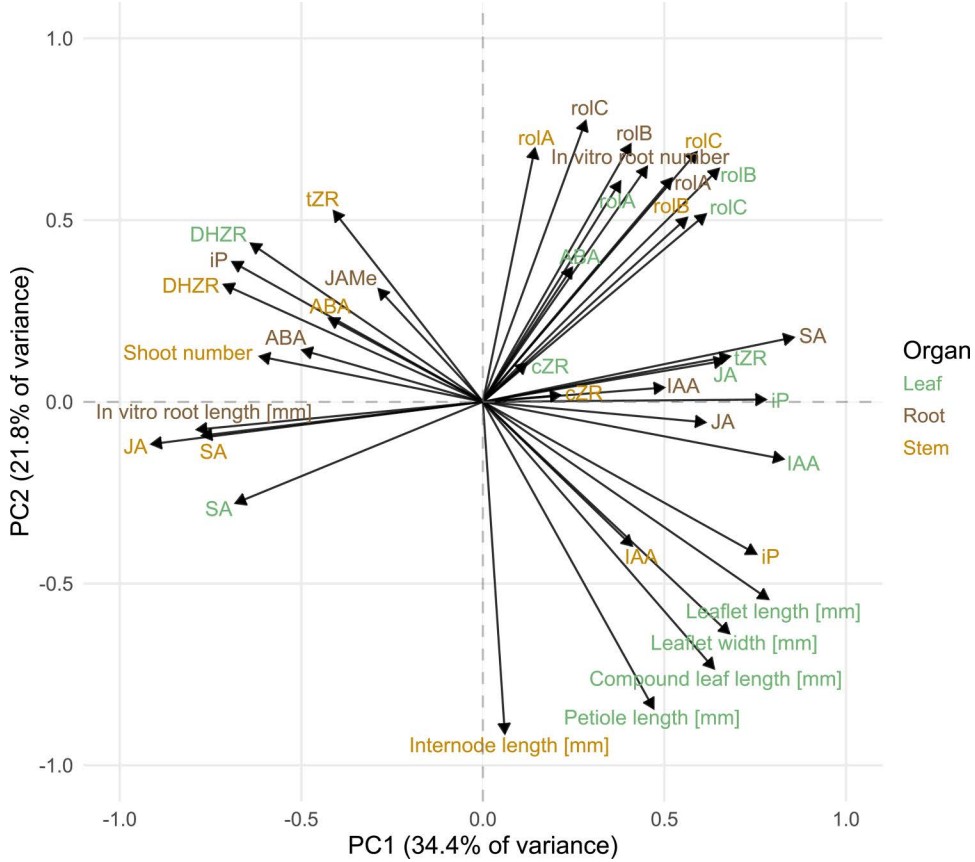

**Fig 5. Principal component (PC) analysis biplot of plant morphological, gene expression, and hormone data.** Each arrow represents a parameter, with the direction indicating the loading direction and the length being proportional to the contribution of that parameter to the principal components. Parameters include morphological traits (internode length, shoot number, compound leaf length, petiole length, leaflet length and width, in vitro root number and length), gene expression data (*rolA*, *rolB*, *rolC*), and hormone levels (IAA, ABA, SA, JA, tZR, DHZR, cZR, iP, and JAMe) measured in different plant organs (stem, leaf, root).

positions of certain hormones, such as SA, JA, iP, and tZR, which were located on opposite sides of PC1, appeared to be strongly organ-specific.

## Discussion

### Species-independent and species-specific traits in Ri plants

Our comparative analysis of rose and potato Ri genotypes revealed several morphological changes characteristic of *rol* gene activity: In both species, the Ri genotypes presented significantly shorter internodes, smaller leaves, and enhanced in vitro rooting. However, the extent of these changes varied considerably among genotypes, likely because of differences in integration sites and/or copy numbers [40,41]. Correlation analyses and PCA further demonstrated that *rolA*, *rolB,* and *rolC* expression levels were most strongly associated with increased root number and reduced internode length and leaflet size. These findings support earlier studies on single species that reported reduced height and smaller leaves in plants overexpressing *rol* genes. Multiple studies have demonstrated similar morphological effects of *rol* gene overexpression in various plant species. In *Nicotiana tabacum*, constitutive expression of *rolA* under the 35S promoter resulted in stunted growth, dark green wrinkled leaves with altered length-to-width ratios, and shortened internodes [11]. Similarly, *rolC*

overexpression in both tobacco and potato (*Solanum tuberosum*) led to dwarfism, and in tomato plants, *rolB* gene expression led to decreased apical dominance and reduced internode length [42]. Studies on several plant species transformed with wild-type T-DNA have likewise reported altered root, internode and leaf parameters, which are listed in the review of Desmet et al. (2020) [2].

We also identified several species-specific responses to the integration of natural T-DNAs. For example, distinct differences in hairy root traits and hormone profiles were evident between rose and potato Ri genotypes. For instance, while stronger root growth, greater root numbers, and greater root lengths were observed in all the Ri genotypes than in their respective wild-type genotypes, the characteristic hairy root phenotype, characterized by a greater number of root hairs, was uniquely observed for the roots of the potato Ri genotype Ds3 but not in any of the rose Ri genotypes. Furthermore, contrasting patterns in terms of plant hormone concentrations were observed: iP and IAA were present at lower concentrations in the leaves and stem tissues of rose Ri genotypes but at higher concentrations in those of potato Ri genotypes. These differences suggest variations in how different plant species perceive, integrate, or respond to the altered hormonal balance induced by *rol* gene expression. The strength of the present study lies in the parallel analysis of these phylogenetically distinct species with the same methodology under identical laboratory conditions. This ensured that the observed species-specific traits could not be attributed to variations in experimental design. Moreover, the consistent differences observed between potato and rose species across multiple Ri genotypes within each species suggest that they are not merely a consequence of specific T-DNA integration sites but rather reflect species-level responses to *rol* gene activity.

Despite markedly improved in vitro root formation, compared with wild-type plants, acclimatized Ri plants presented reduced shoot numbers and reduced root biomass—an observation that contrasts with some prior reports [2]. Similarly, although the potato tuber shape was altered in line with observations by Ooms et al. (1986) [43], the total tuber mass was significantly lower in Ds1, Ds2 and Ds4 than in the wild-type. These results raise concerns about the usefulness of Ri technology application in crops, where the focus is on yield, whereas this does not pose a problem for ornamental applications, such as potted roses.

Finally, a first survey of root hair traits in Ri genotypes revealed no consistent increase in root hair density or length in roses, contradicting the hypothesis that plants regenerated from hairy roots would also have hairy root systems. Visual inspection of in vitro rooted Ri shoots revealed similar results, where only Ds3 exhibited a pronounced root hair morphology. These results complement our observations of several potato hairy root cultures, whose root hair density increased only in some cases. Interestingly, a consistent and significantly larger root body area without root hairs was detected in all the Ri genotypes. This suggests a *rol*-mediated increase in vascular or cortical radial growth, similar to observations in transformed Ri walnut [44].

### Altered auxin and cytokinin homeostasis in Ri genotypes

Hormonal profiling across organs revealed a species-independent pattern of elevated cZR, tZR and DHZR in aerial tissues, each of which was positively correlated with *rol* gene expression in several cultivars. Similar reports linking T-DNA gene expression to increased cytokinin content have been published by Alcalde et al. (2022) [45] in *Centella asiatica* hairy roots and Schmülling et al. (1993) [13] in transgenic tobacco and potato plants. This pattern aligns with the known function of *rolC* as a cytokinin β-glucosidase that catalyzes the release of active cytokinins [9].

Similarly, auxin dynamics were influenced: *aux2* gene ($T_R$-DNA) expression was negatively correlated with the leaflet size and leaf IAA content. This is surprising because the *aux1* and *aux2* enzymes are known to facilitate the conversion of tryptophan to indole-3-acetamide and IAA [8,46,47]. Studies on the influence of T-DNA on auxin levels have reported diverse and sometimes contradictory effects. There are reports of auxin reduction due to the involvement of $T_L$-DNA: Gartland et al. (1991) [10] demonstrated that *rolC* expression decreased free IAA levels in the transformed roots of *Solanum dulcamara*. Similarly, Dehio et al. (1993) [11] reported a 40% reduction in the free auxin content in young transgenic tobacco seedlings. Furthermore, increased sensitivity to auxin was reported in tissues with $T_L$-DNA genes, which further

complicates the identification of the role and effect of the $T_R$-DNA. For example, the *rolA* gene is known to decrease the auxin content while also increasing sensitivity to it through the manipulation of signal transduction through the plasma membrane in tobacco leaf plasma membrane vesicles [48]. Studies on *Nicotiana tabacum* mesophyll protoplasts have demonstrated that individual *rol* genes can increase auxin sensitivity, with *rolB* being the most powerful among the *rol* genes in enhancing cellular responses to auxin [49]. The protein of the gene *ORF8* (located on the $T_L$-DNA) was similar to the rolB protein in its N-terminal domain and may modulate auxin responsiveness in host cells. ORF8 might also increase IAA levels because of its ability to facilitate the conversion of tryptophan to IAM [16]. Finally, in a tobacco protoplast β-glucuronidase assay, *rolB* gene expression was shown to be enhanced by different auxin-type growth regulators, indicating the presence of a feedback system [50]. Therefore, the negative correlation of the *aux2* gene and IAA in leaves might originate from a complex interplay of several genes on the $T_L$- and $T_R$-DNA that potentially interact with auxin-conjugate release mechanisms. Unfortunately, the number of $T_R$-containing genotypes in this study was too small to draw firm conclusions about the role of the genes on the $T_R$-DNA in shaping the Ri phenotype.

### Altered stress hormone levels suggest physiological trade-offs

All the Ri genotypes exhibited altered levels of stress-related plant hormones—JA, SA and ABA—in at least one or more plant organs. This increase in hormone content can be attributed to T-DNA integration and not to the environment, since all Ri genotypes were compared to the respective wild-type plants, which were grown under the same conditions. In general, organ-specific hormone accumulation occurs because the levels of stress hormones such as JA and SA are precisely controlled not only by their production but also by how they are transported, perceived, and deactivated. This mechanism is used by plants to activate tailored defense mechanisms depending on the specific stress and its duration. Different plant species also possess unique genetic programs and tolerances to cope with environmental challenges [51–54]. The shifts in JA and SA levels, along with changes in leaf color (chlorosis in Am1/Ds3; darkening in Am4), suggest the activation of senescence, defense pathways, or potential oxidative or light stress [55–57]. In particular, compared with the Am wild-type, the Am4 Ri genotype displayed darker leaves with potentially higher chlorophyll contents and a stronger reduction in JA in leaves. Comparable results have been obtained in rice and *Arabidopsis*, where increased JA levels were linked to increased chlorophyll degradation [58,59].

Elevated ABA levels might further indicate water deficit signaling or stomatal adjustments, which is consistent with reports about reactions of Ri plants to osmotic stress [1]. Recently, a naturally integrated *rolD*-like gene in sweet potato was shown to be involved in the response to abiotic stresses such as heat, cold, salt, drought, high light, and UV radiation [60], supporting the role of *rol* genes in stress adaptation mechanisms. Alcalde et al. (2022) [45] demonstrated that in *Centella asiatica* hairy roots, ABA was critical in defining morphological traits and closely linked to *rol* gene expression. Veremeichik et al. (2022) [19] reported that the constitutive overexpression of *rolB* in *Arabidopsis thaliana* transgenic plants led to the activation of the SA signaling system, resulting in increased flavanol accumulation and increased drought tolerance through the cooperative action of the SA and reactive oxygen species (ROS) pathways. Similarly, Shkryl et al. (2024) [20] reported that heterologous expression of *rolB* and *rolC* in *A. thaliana* increased the accumulation of SA and MeJA and, to a lesser extent, elevated ABA levels.

Although altered stress hormone profiles may affect plant performance, such as biomass or tuber yield, reports exist concerning the advantages of Ri plants under stress conditions. With respect to drought tolerance, Chen et al. (2025) [61] demonstrated that Ri oilseed rape maintained higher hydraulic integrity under severe soil drought. Chen et al. (2024b) [1] further investigated osmotic stress resilience and reported that compared with wild-type plants, Ri oilseed rape genotypes presented less severe leaf wilting, greater stomatal conductance, and more stable leaf transpiration rates. Enhanced antioxidant capacity might represent another key advantage, as plants such as *Withania coagulans* and *Artemisia carvifolia* expressing the *rolA* gene have demonstrated significant increases in antioxidant potential compared with their untransformed wild-type counterparts [62,63]. Shkryl et al. (2022) [64] reported the transcriptional regulation of enzymes involved in ROS metabolism and abiotic stress tolerance in

*rolC*-transformed cell cultures. Biotic stress tolerance can potentially be improved as well, as demonstrated by the enhanced tolerance to fungal pathogens in Ri *Ajuga bracteosa* and *rolA/rolB*-expressing *Solanum lycopersicum* [42,65,66].

## Conclusions, limitations and future research directions

Our study confirms that Ri technology produces consistent morphological changes across plant species, including shorter internodes, smaller leaves, and enhanced rooting, with these effects being strongly linked to *rol* gene expression levels. Moreover, there appear to be species-specific responses to T-DNA gene expression for cytokinin and auxin contents and, potentially, for root hair morphology. In general, Ri genotypes experienced altered hormone homeostasis, particularly elevated levels of cytokinins, along with the activation of stress-related pathways, as indicated by changes in stress hormone concentrations and leaf coloration.

There are several considerations for future investigations of the analysis of Ri genotypes: since we collected morphological, gene expression, and hormone data from separate batches of plants in different experiments, we had to use average values for each genotype in our correlation analyses, which likely masked individual differences. Nevertheless, the correlations found may be especially stable and meaningful. In future studies, all measurements should be taken from the same plants to improve statistical power. In addition, the number of $T_R$ genotypes should be increased for more conclusive results regarding the effect of the $T_R$ DNA genes. GA profiles and more types of auxin conjugates should be included in future studies, which will aid in obtaining a more complete picture of hormonal changes and their effects on morphology. Expression of *rol* genes is well known to negatively correlate with GA levels and shorter internodes in tobacco plants overexpressing *rolA* [11,15].

Furthermore, experimental studies investigating the effects of T-DNA gene expression on hormone metabolism might also include additional expression of transgenes that upregulate or downregulate the expression of certain hormone signaling components in plants. This could clarify the impact of T-DNA gene expression and hormonal crosstalk to a certain extent and could be facilitated by the use of mutant gene banks with knockouts of genes known to affect hormone pathways. Another strategy could comprise co-transformation with binary plasmids that simultaneously integrate wild-type T-DNA and RNAi constructs to silence certain genes of interest.

To better understand stress responses and fully assess the potential of Ri plants for horticultural and agricultural applications, future research should integrate various parameters to characterize stress physiology. This includes quantification of reactive oxygen species and antioxidants and analysis of the relative agronomic performance of Ri genotypes under stress conditions. Investigations utilizing tools such as chlorophyll fluorescence, membrane integrity testing, and pathogen challenge assays are crucial for evaluating photosynthetic performance and stress-related metabolites under both pest/disease and environmental stresses. This would help determine whether the *rol*-driven growth changes come at the cost of lower plant health or yield and whether stress resilience needs to be improved in parallel.

In addition to providing physiological insights, this study has shown the Ri technology's great potential for practical applications in plant breeding in both agriculture and horticulture. This characteristic of reduced growth might facilitate the development of new rootstock genotypes for fruit trees or the production of own-rooted scions. In ornamental horticulture, this trait could reduce the reliance on chemical growth retardants for plants such as chrysanthemums, promoting more sustainable cultivation. Beyond compactness, stronger root systems might lead to increased water uptake efficiency. Furthermore, Ri technology might contribute to increased tolerance to abiotic stresses, such as drought stress, while altered stress hormone metabolism might also lead to changes in tolerance or susceptibility to pathogens, which should be considered in the development of disease-resistant cultivars.

## Supporting information

**S1 File. Raw Data. Measurements of morphological, expression and hormone data.** In vitro roots, internodes, shoots, leaf parameters, dry masses, root hair parameters, primer efficiencies, Ct values, hormone analysis matrix, hormone concentrations.
(XLSX)

**S2 File. Data Summary. Statistics of the analyzed data.** Overview of statistical models, means, standard deviations, n numbers, p values, and ratios to the reference wild-type (except for expression data) of morphological, expression and hormone data, as well as results of organ and cultivar dependent correlations.
(XLSX)

**S3 File Materials and Methods. Supplementary Material and Methods.** Details about transformation and regeneration of Ri genotypes, overview of experiment chronology, Material and Methods of dry mass, root hair and plant hormone analysis, and primer overview.
(DOCX)

**S4 File. Results. Supplementary results.** Graphical presentations of morphological, gene expression and hormone data, gene expression ANOVA table, correlation tables of significant correlation across all Ri genotypes, correlations with at least one significant correlation in one cultivar, correlations with the same trend across all three cultivars.
(DOCX)

**S5 File. RStudio Scripts. RStudio scripts for statistical analysis and creation of graphics.**
(ZIP)

## Acknowledgments

The authors gratefully acknowledge the technical staff of the Section Reproduction and Development and the Section Molecular Plant Breeding for their assistance with cultivation of plant material, cDNA synthesis, and conducting parts of the quantitative real-time PCR analyses.

## Author contributions

**Conceptualization:** Philipp Rüter, Tim Thomsen, Traud Winkelmann, Thomas Debener.

**Data curation:** Philipp Rüter, Tim Thomsen, Piotr Waligórski, Ewa Surówka, Jacek Kęsy.

**Formal analysis:** Philipp Rüter, Tim Thomsen, Manh Hung Doan.

**Funding acquisition:** Jarosław Tyburski, Traud Winkelmann, Thomas Debener.

**Investigation:** Philipp Rüter, Tim Thomsen, Jarosław Tyburski, Piotr Waligórski, Ewa Surówka, Jacek Kęsy, Natalia Mucha, Manh Hung Doan.

**Methodology:** Philipp Rüter, Tim Thomsen, Piotr Waligórski, Ewa Surówka, Jacek Kęsy, Traud Winkelmann, Thomas Debener.

**Project administration:** Jarosław Tyburski, Traud Winkelmann, Thomas Debener.

**Software:** Philipp Rüter, Tim Thomsen.

**Supervision:** Traud Winkelmann, Thomas Debener.

**Visualization:** Philipp Rüter.

**Writing – original draft:** Philipp Rüter.

**Writing – review & editing:** Philipp Rüter, Tim Thomsen, Jarosław Tyburski, Piotr Waligórski, Ewa Surówka, Jacek Kęsy, Traud Winkelmann, Thomas Debener.

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
