## [Decision Letter · Decision Letter 0]

18 Dec 2025

Dear Dr. Rüter,

We look forward to receiving your revised manuscript.

Kind regards,

Muhammad Abdul Rehman Rashid, PhD

Academic Editor

PLOS One

Journal Requirements:

Reviewers' comments:

Reviewer's Responses to Questions

**Comments to the Author**

1. Is the manuscript technically sound, and do the data support the conclusions?

Reviewer #1: Yes

Reviewer #2: Yes

Reviewer #3: Yes

Reviewer #4: No

2. Has the statistical analysis been performed appropriately and rigorously?

Reviewer #1: Yes

Reviewer #2: Yes

Reviewer #3: Yes

Reviewer #4: Yes

3. Have the authors made all data underlying the findings in their manuscript fully available?

Reviewer #1: Yes

Reviewer #2: Yes

Reviewer #3: Yes

Reviewer #4: Yes

4. Is the manuscript presented in an intelligible fashion and written in standard English?

Reviewer #1: No

Reviewer #2: Yes

Reviewer #3: Yes

Reviewer #4: Yes

Reviewer #1: Dear authors

Comments considered must be required for publication. The present article, titled Morphological and hormonal diversity in rose and potato Ri genotypes: A comparative study, is acceptable, but after the requested items.

Best regards,

Reviewer #2: The paper is a nice combination of different analysis on Ri plants and is valuable for everyone working with the technology. As far as I am aware of, the technology is always described as being a non-GMO technology as long as natural strains are being used. As such, to avoid confusion, I would like to have the paragraph (lines 50-57) removed.

Regarding the Gene expression analysis:

- According to the MIQE guidelines, you have to write RT-qPCR instead of qRT-PCR.

- have you also included a NTC? And how sure are you about the fact that there is no gDNA contamination? I don't see the use of noRT samples and for bacterial (T-DNA) genes, the development of intron-spanning genes is not possible.

Regarding the plant hormone analysis:

samples of 100-200 mg coming from 4-7 replicates were taken. This sounds as a huge variation to me. Could this have had an impact on the outcome?

Line 506 - typo: correlation auf... should be of

Reviewer #3: This is a valuable manuscript that investigates the effects of rol genes on morphological characteristics and endogenous composition. The manuscript is clearly written and well structured. I have only a few minor comments, which are listed below.

-In the caption of Table 1, please also specify the R. rhizogenes strain that was used.

-If possible, please include additional information on the copy number of the introduced genes, as gene copy number can influence plant characteristics.

-Several data sheets are mentioned in the manuscript; please ensure that all such references consistently refer to the appropriate supplementary data sheets (include S number).

Typographical corrections:

• Line 506: “auf” should be corrected to “of”.

• Line 559: “Since” should not be capitalized.

Reviewer #4: Dear authors,

The article's topic is very interesting. However, the work itself is very crude, both in its presentation and in the presentation of the most significant data.

1. In the introduction, the influence of auxins on the expression of rol genes was ignored, although it is an important factor in the mutual regulation of their expression. The work was stated to be devoted to the study of hormone homeostasis, but the results do not confirm this.

2. In the section on expression in the Materials and Methods chapter, it's unclear whether melting curve analysis was used for each sample or only for primer testing. This is an important consideration when analyzing transgenes.

3. The most important stages of hormone analysis are included in the appendix. The instrument used should be included in the body of the article. Why was such a complex, multi-step sample preparation method used? This type of sample preparation is difficult to standardize. Furthermore, the detection limit of hormones must be specified in ng, as calculated for the samples, not in Fmol. The device used does not have sufficient resolution to detect differences in hormonal profiles. Multi-step sample preparation allows for the concentration of hormones for detection, but may impact the accuracy of quantitative analysis.

4. In this work, IAA is of greatest interest because it is involved in the mutual regulation of the expression of rol genes, especially the rolA and rolB genes. Unfortunately, it's impossible to establish relationships between IAA content and rol gene expression in roots and shoots in this study. Material of varying ages and origins was used. Am I correct in my understanding?

5. Gene expression analysis is only provided in the supplement. It's very difficult to understand what can be compared with what. Is it possible to compare expression in shoots and roots? Or are they also of different ages and origins? Is it possible to compare the expression of the rola, b, and c genes within a single sample?

6. It seemed that the authors were not confident in this data or for some reason could not analyze it and determine the patterns.

7. Since the authors draw conclusions about IAA and JA (Table 2), it is necessary to place these two hormones in a histogram separate from the heatmap. Since the authors draw conclusions about expression (Table 2), it is necessary to display these data in the histogram in the results.

8. I was very surprised that the discussion focused on the effects of natural transformants, while the work was about direct transgenesis. And if data on rol gene expression and hormone levels were obtained, they need to be discussed in light of similar publications.

advisory comments

9. If the authors doubt the obtained data (hormone levels and rol gene expression) and have difficulty interpreting them, I would suggest retaining only the morphometric data. The hormone levels and rol gene expression should be refined. The expression analysis should be conducted more thoroughly and on a homogeneous sample (age and origin).

10. It would be very interesting if IAA content and expression were measured in samples of the same age to identify patterns. IAA is most effectively measured using ELISA. Using diethyl ether-based concentration, sample preparation is simpler than that used in this study. And the method is significantly more sensitive. I don't insist on this for this job, but it might be interesting in the future.

11. It's possible that if the data provided is incorrect, data conflicts may arise in your or others' future research. Since this topic is very interesting and has real potential for practical application, I strongly recommend not rushing into publishing data if the authors themselves are not satisfied with it.

Minor comments:

• Carefully check the formatting of the article in accordance with the requirements of the journal. In addition, the work requires language and professional editing.

• When mentioning a plant in the title, abstract, keywords, and at the first mention in the introduction, the full Latin name must be given (with author). When first mentioned in an introduction, the family must be indicated.

Best regards

**Do you want your identity to be public for this peer review?** For information about this choice, including consent withdrawal, please see our For information about this choice, including consent withdrawal, please see our Privacy Policy .

Reviewer #1: **Yes:** Zohreh HajibaratZohreh Hajibarat

Reviewer #2: No

Reviewer #3: No

Reviewer #4: No

---

## [Author Response · Author response to Decision Letter 1]

23 Feb 2026

Editor Muhammad Abdul Rehman Rashid

• First, we would like to thank the editor for accepting this manuscript to be reviewed by the scientific community. We have carefully reviewed the guidelines and ensured that the manuscript now meets PLOS ONE’s style requirements. Specifically, we have corrected the supplementary file names by replacing blanks with underscores ("_") and have adjusted all headings to the correct format.

• We appreciate the emphasis on reproducibility and open science. We now have added a supplementary zip-file “S5 RStudio Scripts”, in which all author-generated code is made available.

• We apologize for this discrepancy. We have updated the Funding Information section to ensure it is accurate, with the information below:

German funding:

• full name of the founder: Deutsche Forschungsgemeinschaft (DFG)

• grant number: WI 2002/6–1

• link: https://www.dfg.de/

Polish funding:

• full name of the founder: National Centre for Research and Development (NCBR)

• grant number: SUSCROP/II/RootsPlus/02/2021

• link: https://www.gov.pl/web/ncbr

Suscrop funding:

• full name of the founder: European Union’s Horizon 2020 research and innovation program

• grant number: 771134

• link: https://research-and-innovation.ec.europa.eu

Reviewer #1: Dear authors

Comments considered must be required for publication. The present article, titled Morphological and hormonal diversity in rose and potato Ri genotypes: A comparative study, is acceptable, but after the requested items.

• We thank the reviewer for their positive assessment of our manuscript and for confirming that it is appropriate for publication. We have addressed the requested corrections as stated below.

This MS appropriate for publication. However, some correction should be performed.

Abstract

-The abstract was inadequate, and further results were furnished.

• We have revised the abstract to include specific quantitative results from our experiments, providing a more detailed summary of our findings.

Introduction

- How were these organisms chosen, and why were other possible species deemed inappropriate for sampling in this work?

• We appreciate this question. As outlined in the introduction, we selected rose and potato because they represent phylogenetically distinct clades (rosids vs. asterids) and possess distinct physiological habits (woody vs. herbaceous). This makes them ideal candidates to evaluate whether the changes induced by wild-strain T-DNA integration are species-dependent or -independent. We have expanded the introduction (line 77-80) to clarify this rationale further.

Material and methods

In vitro root measurements:

- This section should be written briefly.

• We have reviewed this section and shortened it according to your comment. We believe the current level of detail is necessary to ensure the experiments can be accurately reproduced by other researchers.

T-DNA gene expression analysis

- Please elucidate this more scientifically and concisely.

• We have revised this section to be more precise. We have added specific details regarding the digestion of genomic DNA (gDNA) and the controls used to detect potential contamination. Furthermore, we have included additional information regarding the validation of primer specificity via melting curve analysis.

Results

Some sentences need to revision. Please check the entire text. Displayed are ratios which were calculated by dividing the mean of the Ri genotype by the mean of the respective wild-type in page 9 (183-185). The sentence require general revisions in page 17 (lines 344-345).

• We have thoroughly checked the entire text and improved the sentence structure, including the specific lines mentioned, to enhance readability and clarity. In addition, we have commissioned the American Journal Experts to enhance the readability of the manuscript.

Conclusion

This section is poor and needs to improve for your results. Further, there are no comparison between two species.

• Thank you for your comment, which we carefully thought of. Please find an explanation of our motivation and line of arguments for the conclusions section in the following paragraph. We hope that you can agree.

• All results, including the comparison between the two species, are summarized in the first paragraph of the conclusion. While direct comparisons of gene expression levels or hormone concentrations between species are challenging due to the methods used (different reference genes, different plant material), we have focused on comparing the patterns and correlations of ratios of parameters from wild-types versus Ri genotypes across both species. We believe, that this is the most crucial aspect of the manuscript, is methodically sound and enabled us to observe the overall species-de- and -independent impact of wild-strain T-DNA on the phenotype and hormone profiles of Ri genotypes. Additionally, we have elaborated on optimizations for future experimental designs and the broader impact of this study.

Reviewer #2: The paper is a nice combination of different analysis on Ri plants and is valuable for everyone working with the technology. As far as I am aware of, the technology is always described as being a non-GMO technology as long as natural strains are being used. As such, to avoid confusion, I would like to have the paragraph (lines 50-57) removed.

• We sincerely thank the reviewer for their kind words about the value of our study. We agree with the suggestion to avoid confusion regarding the regulatory status. We have removed the specified paragraph and replaced it with a brief remark noting that these plants are generally considered non-GMO when natural strains are used.

Regarding the Gene expression analysis:

- According to the MIQE guidelines, you have to write RT-qPCR instead of qRT-PCR.

• Thank you for pointing this out. We have corrected the terminology to "RT-qPCR" throughout the manuscript to comply with MIQE guidelines.

- have you also included a NTC? And how sure are you about the fact that there is no gDNA contamination? I don't see the use of noRT samples and for bacterial (T-DNA) genes, the development of intron-spanning genes is not possible.

• This is an important point. NTCs were included for all perfomed qPCR runs and can be found in in the supplementary file S1_Raw_Data.xlsx in the sheet “Expression data”. An additional indication was added in the Materials and Methods section of the manuscript to highlight this. The RNA-extraction kit involves digestion of remaining gDNA with DNAseI. After cDNA synthesis, samples were assessed for potential genomic DNA contamination using intron-spanning actin primers. This approach was used to indicate general plant gDNA contamination in the samples assuming that all gDNA had been removed and therefore was not additionally performed for the transgenes. The respective procedure is now included in the manuscript and the actin primer sequences have been added to the supplementary primer list (Supplementary file S3 Materials and Methods, Table B).

Regarding the plant hormone analysis:

samples of 100-200 mg coming from 4-7 replicates were taken. This sounds as a huge variation to me. Could this have had an impact on the outcome?

• We acknowledge the reviewer's observation regarding sample variation. The standard deviations (SDs) are documented in supplementary file S2, sheet "Hormone data". While some variation exists, we applied robust statistical procedures, including log-transformation and residual analysis, to ensure the validity of our linear models. Despite the variation, we were able to identify significant differences and consistent patterns in the hormone heatmap and the correlation analyses. We have discussed in the conclusion that future experimental designs could be optimized to further increase statistical power, but we are confident that the current results support our conclusions.

Line 506 - typo: correlation auf... should be of

• We have corrected this typo, thank you!

Reviewer #3: This is a valuable manuscript that investigates the effects of rol genes on morphological characteristics and endogenous composition. The manuscript is clearly written and well structured.

• We thank the reviewer for this positive feedback and for taking the time to review our work.

I have only a few minor comments, which are listed below.

-In the caption of Table 1, please also specify the R. rhizogenes strain that was used.

• We thank the reviewer for this helpful suggestion. We have added the strain information to the caption of Table 1.

-If possible, please include additional information on the copy number of the introduced genes, as gene copy number can influence plant characteristics.

• We agree that correlating copy number with gene expression would be highly interesting. Unfortunately, establishing a reliable protocol for copy number identification via droplet digital PCR (ddPCR) has proven to be surprisingly difficult for these genotypes. As we are still working on optimizing this method, it was not possible to include this data in the current study.

-Several data sheets are mentioned in the manuscript; please ensure that all such references consistently refer to the appropriate supplementary data sheets (include S number).

• We have double-checked all references to supplementary data. We have ensured that every mention of a data sheet is now accompanied by the corresponding Supplementary File number (e.g., S1, S2) to avoid confusion.

Typographical corrections:

• Line 506: “auf” should be corrected to “of”.

• Line 559: “Since” should not be capitalized.

• We have corrected both typographical errors, thank you!

Reviewer #4: Dear authors,

The article's topic is very interesting. However, the work itself is very crude, both in its presentation and in the presentation of the most significant data.

• We thank the reviewer for finding the topic interesting and for their critical assessment. We have carefully addressed the specific concerns raised to improve the presentation and clarity of our data.

1. In the introduction, the influence of auxins on the expression of rol genes was ignored, although it is an important factor in the mutual regulation of their expression. The work was stated to be devoted to the study of hormone homeostasis, but the results do not confirm this.

• We thank the reviewer for highlighting this important interaction. We have added this information to the dedicated paragraph in the Discussion section detailing how auxins can influence rol gene expression. We also wish to clarify that the primary focus of this study was to observe the impact of the whole, wild-strain T-DNA on several key hormones across two distinct species, rather than to uncover specific regulatory mechanisms. This has not been done so far with the whole, wild-strain T-DNA. Most studies investigated only single rol genes and also single species, sometimes not even in plants (hairy root cultures, protoplasts). We believe this species-independent perspective provides valuable information for breeders and researchers working with Ri genotypes.

2. In the section on expression in the Materials and Methods chapter, it's unclear whether melting curve analysis was used for each sample or only for primer testing. This is an important consideration when analyzing transgenes.

• We apologize if this was unclear. We now have clarified in the Materials and Methods section that melting curves were generated and assessed for all reactions—both during primer testing and during the actual expression analysis—to ensure specificity. We also added information regarding the validation of amplicons via agarose gel electrophoresis.

3. The most important stages of hormone analysis are included in the appendix. The instrument used should be included in the body of the article.

• We agree that the analytical instrumentation should be clearly stated in the main text. Therefore, we have added the details on the LC–MS/MS system to the Materials and Methods section.

Why was such a complex, multi-step sample preparation method used? This type of sample preparation is difficult to standardize.

• The multi-step sample preparation, including SPE clean-up, was deliberately selected for two major reasons: (i) removal of matrix constituents and potential interferents, which is crucial for LC–MS/MS quantification of phytohormones in complex plant extracts; and (ii) concentration of analytes, enabling reliable detection and quantification of low-abundance hormones.

While the Reviewer notes that such sample preparation can be difficult to standardize, in practice SPE-based workflows are widely used and well-established in phytohormone analysis. The applied procedure follows standard, reproducible steps (fixed sorbent type, solvent composition, volumes, washing/elution sequence), and its main goal is to ensure reduced matrix effects and improved analytical performance.

There are hundreds of primary and secondary metabolites present in the extracts at much higher concentrations than phytohormones. These substances cause ion suppression in LC-MS/MS analysis, reducing analyte signal intensity (sensitivity), particularly in electrospray ionization (ESI).

• To address potential concerns regarding quantitative accuracy when using multi-step sample preparation, we emphasize that isotope-labelled internal standards were added at an early stage of extraction. This approach is standard in quantitative LC–MS/MS and compensates for potential losses during extraction, SPE, evaporation and reconstitution steps, as well as for matrix effects in ESI. Therefore, the applied internal standard strategy effectively minimizes any negative impact of complex sample preparation on quantification quality and comparability between samples.

Furthermore, the detection limit of hormones must be specified in ng, as calculated for the samples, not in Fmol.

• We agree that detection limits should be interpretable in the context of sample material. We reported LOD/LOQ in fmol, which is a common unit in LC–MS/MS quantitative analysis and directly reflects the instrument’s absolute sensitivity. However, to align with the Reviewer’s suggestion, we have converted the values to express detection limits as mass equivalent (ng) per sample, based on the actual sample mass/extraction volume used. This will make the analytical performance easier to interpret.

The device used does not have sufficient resolution to detect differences in hormonal profiles. Multi-step sample preparation allows for the concentration of hormones for detection, but may impact the accuracy of quantitative analysis.

• We respectfully clarify that the applied method relies on triple quadrupole LC–MS/MS in MRM mode, which is specifically designed for high selectivity and sensitivity in targeted quantification. In this type of analysis, the key determinant is not chromatographic “high resolution” in the sense of HRMS, but rather: chromatographic separation sufficient to avoid co-elution, and the selectivity of MRM transitions combined with retention time matchi

---

## [Decision Letter · Decision Letter 1]

12 Mar 2026

Morphological and hormonal diversity in rose (*Rosa hybrida* L.) and potato (L.) and potato (*Solanum tuberosum* L.) Ri genotypes: A comparative studyL.) Ri genotypes: A comparative study

PONE-D-25-62600R1

Dear Dr. Rüter,

We’re pleased to inform you that your manuscript has been judged scientifically suitable for publication and will be formally accepted for publication once it meets all outstanding technical requirements.

Kind regards,

Patrick Goymer

Staff Editor

PLOS One

Additional Editor Comments (optional):

Reviewers' comments:

Reviewer's Responses to Questions

**Comments to the Author**

Reviewer #1: All comments have been addressed

Reviewer #3: All comments have been addressed

Reviewer #4: (No Response)

2. Is the manuscript technically sound, and do the data support the conclusions?

Reviewer #1: Yes

Reviewer #3: (No Response)

Reviewer #4: No

3. Has the statistical analysis been performed appropriately and rigorously?

Reviewer #1: Yes

Reviewer #3: (No Response)

Reviewer #4: No

4. Have the authors made all data underlying the findings in their manuscript fully available?

Reviewer #1: Yes

Reviewer #3: (No Response)

Reviewer #4: Yes

5. Is the manuscript presented in an intelligible fashion and written in standard English?

Reviewer #1: Yes

Reviewer #3: (No Response)

Reviewer #4: Yes

Reviewer #1: (No Response)

Reviewer #3: (No Response)

Reviewer #4: Unfortunately, I can't say the authors have made any significant changes. Overall, the work appears to be quite publishable based on its formal features. However, it would be better to combine the chapter on qPCR with the next one. For a full results chapter, objective results are insufficient. Essentially, the only meaningful result of the work is morphometry. Molecular and biochemical analyses are uninformative. Hormonal analysis does not allow for intra- or intergroup comparisons. Therefore, it will not be of interest to specialists in this field.

**Do you want your identity to be public for this peer review?** For information about this choice, including consent withdrawal, please see our For information about this choice, including consent withdrawal, please see our Privacy Policy .

Reviewer #1: **Yes:** Zohreh HajibaratZohreh Hajibarat

Reviewer #3: No

Reviewer #4: No

---

## [Editor Report · Acceptance letter]

PONE-D-25-62600R1

PLOS One

Dear Dr. Rüter,

I'm pleased to inform you that your manuscript has been deemed suitable for publication in PLOS One. Congratulations! Your manuscript is now being handed over to our production team.

Kind regards,

on behalf of

Dr Patrick Goymer

Staff Editor

PLOS One